# Modeling the Brand Equity and Usage Intention of QR-Code E-Wallets

Faten Aisyah Ahmad Ramli [1], Muhammad Iskandar Hamzah [2,*], Siti Norida Wahab [2] and Rishabh Shekhar [3]

[1] Faculty of Business and Management, Universiti Teknologi MARA Cawangan Kelantan, Machang 18500, Kelantan, Malaysia
[2] Faculty of Business and Management, Universiti Teknologi MARA Cawangan Selangor, Bandar Puncak Alam 42300, Selangor, Malaysia
[3] Symbiosis Centre for Management Studies, Symbiosis International (Deemed University), Mouza Wathoda, Nagpur 440008, Maharashtra, India
* Correspondence: iskandarh@uitm.edu.my

**Abstract:** The proliferation of digital payments has paved the way for the greater use of E-wallets or mobile payments in over-the-counter (OTC) retail transactions. Nevertheless, given its economic and accessibility benefits over NFC forms of mobile payment, relatively little is known about QR-code E-wallet (QREW) adoption from the consumer–brand relationship perspective. The study aims to address this knowledge void by augmenting brand equity elements (perceived value, brand image, and brand awareness) to comprehensively analyze consumers' QREW usage intention in the OTC retail environment. A structural equation modeling analysis was performed on 305 consumers in the greater Klang Valley, Malaysia. The empirical findings suggest that brand awareness positively affects QREW usage intention and mediates the effects of both perceived quality and brand image on the outcome. Moreover, the results reveal a serial mediation effect involving all of the examined factors. Theoretically, this study supplements the literature on mobile payments from the consumer–brand relationship view, in which the predictive nature of brand equity factors is examined separately. In practical terms, considering that the Malaysian market QREW is in a relatively early growth stage, the findings should offer QREW providers insights into how to capitalize on brand equity mechanisms for attracting consumers to utilize their offerings.

**Keywords:** mobile payment; e-wallet; contactless payment; QR-code; brand equity; perceived quality; brand awareness; brand image; fintech adoption; cashless society

## 1. Introduction

Fintech, short for "financial technology", is a growing innovation sector that aims to improve the effectiveness, convenience, and security of traditional financial services by incorporating cutting-edge IT into them. One area of retail that pays a lot of attention to fintech is the electronic wallet (E-wallet) or mobile payment systems. There are a number of synonyms for E-wallet that may be found in the literature. These include mobile payment (m-payment), mobile wallet (m-wallet), virtual wallet, and digital wallet [1]. Using an E-wallet eliminates the need to exchange banking information for monetary transactions or to physically carry banknotes and coins, hence making it a convenient digital payment solution when shopping in physical stores.

In Malaysia, there are 53 different E-wallet options, making up a significant portion (19%) of the local fintech industry [2]. A majority of these offerings capitalize on the QR-based E-wallet (QREW) format as opposed to the near-field communications (NFC) interface due to its cheaper implementation for both merchants and customers. Due to the high levels of competition in this industry, there is now an excess of QREW services on the market, all vying for the same share of a niche that is already well fulfilled by credit and debit cards [3]. Further complicating matters is the fact that debit card usage is on the

rise because of their convenience and the attractive special offers from banks, which may replace QREW as consumers' potential preferred payment option. In the end, consumers may prefer this tried-and-true technology due to its familiarity and reliability, as the use of QREWs is still in its infancy domestically [4].

These issues highlight the relevance of the research involving branding and technology adoption. With the E-wallet competition heating up with new players entering the market, and unfit E-wallet providers exiting, expressive branding relationships appear to be the key for E-wallet providers to maintain a sustainable brand [5]. The extant literature reveals that customers' adoption of newly offered technology-mediated services is largely influenced by variables related to technology acceptance models (e.g., technology acceptance model or TAM, unified theory of acceptance, and use of technology or UTAUT) rather than the consumer–brand relationship experiences.

In this regard, consumer-based brand equity (CBBE) serves as an indicator of how well a brand succeeds in establishing a productive consumer–brand relationship endeavor. Hence, CBBE functions as a key source of competitive advantage, and many studies have empirically validated CBBE-based frameworks for technology-driven services [6–8]. The practicality of the CBBE strategy enables QREW brands to rethink their positioning in the minds of consumers in order to overcome the problem of overcrowded competition that leaves consumers with too many options to choose from [3]. As customers frequently evaluate brands through both cognitive and affective lenses, this paradigm shift entails the development of a distinct, memorable, and recognized brand that entices consumers to adopt the QREW technology [7].

The pursuit of branding supremacy appears to be relevant considering the rise in popularity of QREW in third-world countries, which are currently in the early phases of its adoption. In this regard, the affordability of QREWs has led to their rapid and broad acceptance in emerging economies like Turkey, India, and Indonesia. In these markets, QREWs have gained more popularity than NFC-based E-wallets for economic reasons [9]. The NFC interface has the disadvantages of requiring a membership to bank services and expensive hardware investments (e.g., terminals for merchants, and NFC-compatible smartphones for consumers) [1]. Inadvertently, these elements have made QREWs an attractive OTC mobile payment option, given that it just requires minimal setup, consisting of QREW app/service registration, a camera-equipped smartphone, and a QR-code print display [10].

QREW providers regularly strive to establish their brand recognition by affiliating with other retail and service brands to aggressively promote special deals and discounts in order to increase penetration rates among newcomers. Numerous QREW app interfaces feature distinctive and eye-catching designs (e.g., logo, theme, etc.) and gamification (e.g., challenges and badges) elements that entice users to explore and experiment with their functions [11]. Eventually, as consumers' minds are constantly imbued with these tangible characteristics, brand recognition is gradually nurtured [12]. However, to what degree these branding campaigns encourage customers to embrace QREWs remains unclear [7]. Moreover, little is known about whether the perception of service quality plays a primary role in evoking favorable mental images of QREW brands [13]. These potential linkages have not been subjected to extensive empirical testing, particularly in emerging markets.

In order to address the scarcity of research that identifies brand equity elements as predictors of QREW adoption, we aim to study these factors independently since this would provide a clearer understanding of the cause-and-effect relationship between them. This approach also offers some theoretical novelty to complement the current literature and answers researchers' call for an independent assessment of a specific type of mobile payment adoption [14]. Therefore, we seek to address the following research questions. (1) Does brand awareness lead to QREW adoption? (2) How do CBBE elements affect consumers' QREW adoption?

This study's primary objective is to examine the interrelationships between brand equity components (perceived value, brand image, and brand awareness) and their influence

on consumers' QREW usage intention in the OTC retail context. In addition, we intend to examine a number of indirect pathways that should augment the role of brand equity as an effective mechanism for modeling users' technology utilization from the perspective of the consumer–brand relationship. The originality of this work is exemplified by the modeling of CBBE components as predictors in a disaggregated approach and by analyzing technology-based financial service adoption outside of the well-established technology acceptance domain (e.g., TAM, UTAUT) as recommended by scholars, evidenced by their recent systematic literature and meta-analytic reviews [1,15–17].

## 2. Literature Review

### 2.1. Brand Equity

Consumer-based brand equity (CBBE) refers to the value that a brand holds in the minds of consumers [18]. It is the impression consumers have of a brand based on their interactions and experiences with the brand. Marketing scholars have concurred that CBBE is crucial to competitive advantage since it can impact consumer behavior and purchase decisions. Consumers are more inclined to choose brands with high consumer-based brand equity than those with low consumer-based brand equity [19]. In this regard, CBBE is a major contributor to the long-term financial performance of a brand. Brands with a high level of consumer-based brand equity typically charge premium pricing for their products and services and are more resistant to competition pressures [20]. In addition, a brand with strong consumer-based brand equity might benefit from greater customer loyalty and word-of-mouth marketing.

Keller's [18] brand equity pyramid model is a framework that consists of four levels: brand identity, brand response, brand resonance, and brand performance. The pyramid starts with brand identity, which includes brand elements such as name, logo, and symbol, and progresses to brand resonance, which refers to the emotional and psychological relationship consumers have with the brand. In essence, the pyramid model explains the different aspects of brand equity and the progression of the relationship between a brand and the consumer, starting with the tangible elements of the brand (at the base) and reaching the emotional and psychological connection (at the apex). Meanwhile, Aaker's [20] model of brand equity is an extension of Keller's brand equity pyramid model. Aaker's CBBE model incorporates brand loyalty as a dimension of brand equity and introduces "brand personality" and "brand relationship" as the drivers of brand equity.

Therefore, CBBE can be divided into three core components: brand loyalty, perceived quality, and other proprietary brand assets [21]. Brand loyalty is the probability that consumers will continue to purchase a brand's products or services in the future [19]. Perceived quality refers to consumers' perceptions of a brand's products or services' overall excellence or superiority [22]. Other proprietary brand assets include brand awareness, brand image, and other intangible elements that contribute to overall brand equity [23,24]. In the context of QR-code mobile payments, CBBE can play a significant role in determining the success of a brand's mobile payment offerings [25,26]. Specifically, perceptions of quality, brand awareness, and brand image determine consumers' continuous adoption of QREWs. The following sections elucidate the inter-relationships among these elements in forming the main hypotheses of the study.

### 2.2. Brand Awareness and QREW Usage Intention

Brand awareness describes the prominence of the brand in the consumer's mind, which is reflected by the consumer's ability to recognize the brand amongst other competing alternatives [27]. Brand awareness can influence consumers' attitudes and perceptions since it helps consumers to distinguish mainstream brands and enables their decision-making process in view of the various options available in the market. When consumers are familiar with a brand, they are more likely to develop a perspective of what the brand represents, which can subsequently affect their purchasing decisions [28]. In terms of consumers' decision-making process, brand awareness increases consumer consideration while making

purchases. Consumers who have strong brand recognition understand what the brand stands for and what to expect from its products and services [22].

Furthermore, brand awareness is also useful in consumers' attitude formation. How people feel about a brand can be affected by how well they recognize it [21]. For instance, if a brand is always associated with positive attributes like quality and dependability, consumers are more likely to develop a positive attitude or, in simpler words, to like the brand despite the presence of other imitations or look-alikes [29]. Since people develop mental images after interacting with a product or service, these encounters gradually alter their impression of what the brand stands for. From the QREW perspective, the ability to recall a specific QREW brand from the competing brands empowers consumers to recall and compare its positive attributes with that of the competitors [7]. This provides the QREW brand with an edge over its competitors since this positive attitude is likely to transform into consumers' consumption behavior. We, therefore, propose the following hypothesis:

**H1.** *Brand awareness will have a positive influence on E-wallet usage intention.*

### 2.3. Indirect Effect of Perceived Quality on QREW Usage Intention

Perceived quality refers to consumers' perceptions of a brand's products or services' overall excellence or superiority following their recent consumption experience [27]. Perceived quality and brand awareness are critical drivers that contribute to brand acceptance, as it leads to consumer satisfaction and loyalty. When consumers perceive a brand's products or services as high-quality, they are more likely to be satisfied with their purchase and will be more likely to make repeat purchases [30]. In addition, high-quality mobile apps can generate favorable word-of-mouth marketing, which can increase brand awareness and acceptance [31]. Similar advantages are obtainable to QREW providers if they prioritize delivering a high-quality mobile app user experience. When a QREW brand is consistently associated with positive quality attributes, such as responsiveness to a support request and consistency in service delivery, consumers are more likely to establish a favorable impression of the brand [32].

The positive brand recognition of QREWs is also influenced by other product attributes, such as security and usability, which encourage consumers to develop a favorable attitude toward these apps. By using QREWs, consumers are entrusting their personal and financial information to a brand; therefore, security assurance and perception of quality are particularly important factors [6]. As these quality attributes tend to lead to consumers' satisfactory QREWs usage experiences, in tandem, their awareness toward the brand also increases [33]. Thus, this brand recognition promotes QREW adoption by increasing the likelihood that a QREW brand will be considered among the other competitive brands during the payment decision process. Within a relatively nascent mobile payments market like Malaysia, QREW providers routinely conduct aggressive marketing campaigns to reinforce consumers' perceptions of their service quality and brand recognition; eventually, these campaigns gradually increase the brand's acceptance from consumers [34].

Based on the above discussion, the following hypothesis was formulated:

**H2.** *Perceived quality indirectly influences E-wallet usage intention via brand awareness.*

### 2.4. Indirect Effect of Brand Image on QREW Usage Intention

Brand image was defined by Keller [18] as the "*perceptions about a brand as reflected by the brand associations held in consumer memory*". A similar concept of brand image was also put forth by Aaker [20], who viewed brand image as a consumer's interpretation of the marketer and its products and services that take into account their memorable experiences and interactions. Both of their brand image typologies share many common elements, namely user imagery, usage imagery, and product or service attributes. Brand image and

brand associations are used interchangeably as important variables of brand equity within the literature [23,24]. Similarly, brand associations and brand awareness are inseparably connected in creating a unique brand image, consequently contributing to the formation of brand equity [35].

Brand awareness, in turn, promotes brand acceptance by increasing the likelihood that a QREW brand will be considered during the purchase decision process. Additionally, high brand awareness can also lead to favorable evaluation and adoption of mobile payment services [36]. In this regard, the positive brand image associated with a QREW brand can be leveraged to increase consumer awareness and usage. Research has established that when consumers are exposed to positive brand images of a new technology-mediated service, they are more likely to form positive attitudes toward the brand, leading to increased purchase intent and loyalty [33,37]. This can be especially pertinent in the context of mobile payments, as the market is highly competitive, and consumers often have multiple payment options. Thus, a positive brand image can be an effective determinant for QREW brands to differentiate themselves from their competitors and gain market share. Based on these arguments, this study proposes the following hypothesis:

**H3.** *Brand image indirectly influences E-wallet usage intention via brand awareness.*

*2.5. Multiple Serial Mediation Effects of Brand Equity Elements*

Perceived quality is a critical driver of brand acceptance, including QREW adoption, through intangible brand assets, namely brand image and brand awareness. Prioritizing the delivery of a high-quality mobile app user experience can lead to a more favorable brand image, as consumers tend to associate high-quality products or services with positive attributes such as reliability, responsiveness, and consistency in mobile payment service delivery [33]. A favorable brand image can positively influence consumers' attitudes and perceptions toward a brand and ultimately lead to greater brand awareness. When consumers are aware of a brand, they are more likely to consider it when making payment decisions [38]. Likewise, high brand awareness can lead to positive attitudes and perceptions towards a brand, which can positively influence QREW usage intention. Therefore, by delivering a high-quality mobile app user experience, brands can establish a positive brand image and increase brand awareness, which in turn increases the likelihood of QREW adoption.

Similarly, when the quality of a mobile-based service is sub-standard, users will have the impression that the service is too complicated to be useful, thereby creating an image barrier [39]. Consequently, this negative brand image will inhibit the use of the mobile application [40]. Therefore, in addition to prioritizing the delivery of a high-quality mobile app user experience and increasing brand awareness, building consumer trust is also crucial for QREW providers to ensure consumers' continuous usage. Trust in a brand can be built through various means, such as transparent communication, consistent performance, and addressing customer concerns. Consumers are more likely to use a QREW service if they trust the brand and perceive it as reliable [41,42]. Therefore, by delivering a high-quality mobile app user experience (e.g., via fortifying consumer trust and ensuring systems' reliability), establishing a positive brand image, and increasing brand awareness, QREW providers can increase the likelihood of continuous usage of their services. The above discussion leads to the following hypothesis:

**H4.** *Perceived quality indirectly influences E-wallet usage intention through the serial mediation of brand image and brand awareness.*

Figure 1 below illustrates our research model, summarizing the hypotheses presented above.

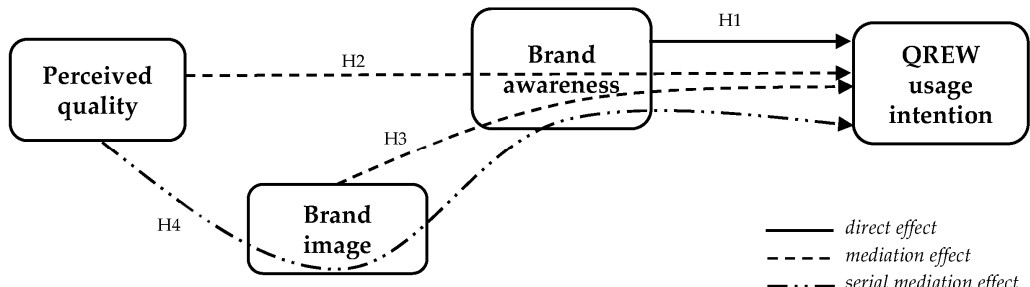

**Figure 1.** Research framework.

## 3. Methodology

### 3.1. Samples and Data Collection Procedures

Through the mall intercept method, respondents were selected via purposive sampling across mainstream malls and convenience stores in Klang Valley, Malaysia. The questionnaire is filled out by the respondents at the exit points of these shopping locations. The mall intercept method provides quick, spontaneous, and honest answers since the respondents' memories about their shopping experience are fresh [43]. In order to be eligible to participate in the survey, participants had to be at least 18 years old and had used a QR-type E-wallet at least once within the previous three months. Respondents were asked to identify a familiar E-wallet brand and assess its user experience in an OTC retail setting. Respondents were asked to rate the E-wallet brand with which they have the most experience, even if they used multiple brands. Out of 375 distributed questionnaires, 305 samples were fit for analysis purposes.

### 3.2. Measures

All variables used in this study were borrowed from existing research and adjusted to be suitable to the E-wallet and OTC retail contexts: perceived quality [12,44], brand image [22], brand awareness [21,44], and usage intention [45]. Two academic experts in technology marketing assisted in evaluating and validating the questions to eliminate those that were unclear, disputed, or suggestive. To assess the latent constructs indirectly, a five-point Likert scale ranging from 1 ("Strongly disagree") to 5 ("Strongly agree") was utilized. Prior to the main survey, a pilot test including 115 respondents revealed an acceptable reliability result.

### 3.3. Analysis Tool and Method Bias

The data analysis was conducted using the partial least square (PLS) path modeling method. By predicting the associations between numerous independent and dependent variables within a structural model, the SmartPLS 3.0 software tool was used to simultaneously test the research hypotheses [46]. Since the latent variables are not directly observable, the indicator (questionnaire item) scores were aggregated to reduce the measurement error and provide a more accurate estimation of the SEM model parameters [47]. To detect any potential bias within the dataset, particularly the potential risk of common method variance, several statistical tests were conducted. Initially, Harman's one-factor test was conducted using the SPSS factor analysis tool. It was found that there is no single general factor that accounts for more than 50% of the covariance among the measure; hence, the common method is not an issue [48]. In another test, a full collinearity test was conducted following [49]. In this test, all the variables were regressed against a common variable, and if the obtained variance inflation factor (VIF) values fell below the 3.3 threshold, there should be no bias from the single-source data. All of the variables have VIF values of less than 3.3 (ranging from 1.24 to 1.59). Hence, common-method bias is not a serious issue in the model.

## 4. Findings

### 4.1. Sample Characteristics

Of the 305 respondents, 61.6 percent were females. The majority of respondents (83.3%) are between the ages of 18 and 39, followed by those aged 40 to 49 (12.1%) and those aged 50 and over (4.6%). A significant portion of respondents (36.7%) hold a bachelor's degree, followed by high school graduates and diploma holders (24.6%), postgraduates (11.3%), and those with other qualifications (24.3%). About two-thirds of the participants were employed (65.9%). In terms of income, the majority of respondents earned RM 2500 or less (64.3%), followed by those earning between RM 2501 and RM 4500 (19.3%), RM 4501 and RM 6500 (9.2%), RM 6501 and RM 8501 (3.9%), and over RM 8501 (3.9%). The sample also reveals that the respondents' preferred E-wallet brands include *Touch n Go* (40.3%), *Boost* (26.2%), and *GrabPay* (23.3%). Table 1 indicates the sample's demographic characteristics.

**Table 1.** Respondents' demographic profile (n = 305).

| Demographic Variable | | n | % | Demographic Variable | | n | % |
|---|---|---|---|---|---|---|---|
| Age | | | | Occupation | | | |
| | 18–23 years old | 93 | 30.5 | | Employed | 201 | 65.9 |
| | 24–29 years old | 79 | 25.9 | | Student | 100 | 32.8 |
| | 30–39 years old | 82 | 26.9 | | Retired | 4 | 1.3 |
| | 40–49 years old | 37 | 12.1 | Income | | | |
| | 50 years & above | 14 | 4.6 | | <RM2500 | 196 | 64.3 |
| Gender | | | | | RM2501–RM4500 | 59 | 19.3 |
| | Male | 117 | 38.4 | | RM4501–RM6500 | 28 | 9.2 |
| | Female | 188 | 61.6 | | RM6501–RM8500 | 12 | 3.9 |
| Education level | | | | | >RM8501 | 10 | 3.3 |
| | Diploma & lower | 75 | 24.6 | Brand | | | |
| | Bachelor's degree | 112 | 36.7 | | Touch n Go | 123 | 40.3 |
| | Postgraduate's degree | 44 | 14.4 | | Boost | 80 | 26.2 |
| | Other qualifications | 74 | 24.3 | | GrabPay | 71 | 23.3 |
| | | | | | Others | 31 | 10.2 |

### 4.2. Measurement Model

The measuring model was evaluated for its reliability and validity. The four standard criteria for evaluating reliability and validity are individual item reliability, construct reliability, convergent validity, and discriminant validity [47]. First, since the latent variables are modeled as reflective, the item loadings of the constructs were observed to ascertain their individual item reliability. The majority of the items exhibit loadings of at least 0.7 s, and the construct reliability of the main variables was measured via the composite reliability (CR) indicator. The CRs for all of the constructs ranged from 0.83 to 0.88 and exceeded the minimum threshold of 0.7. Third, the convergent validity was assessed using AVE. The AVE indicators for all constructs were higher than the 0.5 thresholds (ranging from 0.56 to 0.71). Thus, we concluded that all our constructs had satisfactory convergent validity and reliability. Table 2 presents the values of the loadings, CRs, and AVESs of all of the latent variables. Meanwhile, the complete table of loadings and crossloadings can be referred to in Appendix A.

Finally, we assessed the discriminant validity through the HTMT criterion assessment. The correlation values are less than 0.85 and 0.90 based on this analysis and hence comply with the HTMT.85 [50] and HTMT.90 [51] standards. On the basis of these correlation results (Table 3), it is reasonable to conclude that the measures did not overlap and that discriminant validity has been conclusively established.

**Table 2.** Factor loading, composite reliability (CR), and average variance extracted (AVE).

| Item | Scale | Loadings | CR | AVE |
|------|-------|----------|-----|-----|
| | *Perceived quality* | | | |
| PQ1 | This E-wallet brand offers good service quality. | 0.73 | 0.83 | 0.55 |
| PQ2 | This E-wallet brand is very reliable. | 0.80 | | |
| PQ3 | This E-wallet brand is of better quality compared to other brands. | 0.73 | | |
| PQ4 | This E-wallet brand is trustworthy. | 0.70 | | |
| | *Brand image* | | | |
| BI1 | Some characteristics of this E-wallet come to my mind quickly. | 0.76 | 0.84 | 0.64 |
| BI2 | I can quickly recall the logo of this E-wallet. | 0.86 | | |
| BI3 | I have no difficulty imagining this E-wallet in my mind. | 0.78 | | |
| | *Brand awareness* | | | |
| BA1 | I can recognize this E-wallet brand among competing brands. | 0.74 | 0.83 | 0.63 |
| BA2 | This E-wallet brand is the only brand recalled when I need to make a QR code based cashless transaction. | 0.82 | | |
| BA3 | This E-wallet brand comes up first in my mind when I need to make a QR code-based cashless transaction. | 0.81 | | |
| | *Usage intention* | | | |
| UI1 | I intend to use this E-wallet in the next 2 months. | 0.84 | 0.88 | 0.71 |
| UI2 | I predict I will use this E-wallet in the next 2 months. | 0.87 | | |
| UI3 | I plan to use this E-wallet in the next 2 months. | 0.81 | | |

**Table 3.** Discriminant validity.

| | PQ | BI | BA | INT |
|------|------|------|------|------|
| PQ | | | | |
| BI | 0.406 | | | |
| BA | 0.422 | 0.780 | | |
| INT | 0.399 | 0.384 | 0.527 | |

Note: PQ = perceived quality; BI = brand image; BA = brand awareness; INT = usage intention.

### 4.3. Structural Model

In order to test hypotheses via SEM path analysis, a bootstrap approach with 5000 resamples is utilized to produce t-values for the structural model. We find the value of $R^2$ to be 0.19, respectively, affirming a sufficient explanation of the criterion variable that is attributable to the predictors within the model. The results from the fitness test indicate that our data matches the proposed structural model (chi square = 344.06; SRMR = 0.079); hence, an acceptable model fit was demonstrated [52,53]. Based on lateral collinearity assessment, all of the latent constructs produced VIF values ranging from 1.20 to 1.87. Multicollinearity is not an issue with the model, given that only a VIF value of 5 or greater may indicate a potential collinearity concern [47].

Based on the structural model, a significant positive impact of brand awareness on QRWE usage intention was observed (β = 0.395, *t* = 7.623), thus lending empirical support to H1. To test the mediation hypotheses (H2 to H4), the bootstrapping indirect effect method of Preacher and Hayes [54] was utilized. Preacher and Hayes's [54] indirect effect approach is currently the default method in testing mediation effects as compared to the older Baron and Kenny's [55] method because the latter is no longer relevant due to its requirement for a significant relationship between the independent and dependent variables, ignorance of measurement error, assumption of a single construct mediator, and failure to account for multiple mediators' indirect effects [56].

In hypothesizing H2, we find that brand awareness mediates the effect of perceived quality on QREW usage intention (β = 0.06, *t* = 2.47 CI = [0.02–0.11]). As for H3, we discovered that brand image exerts an indirect influence on QREW usage intention (β = 0.15, *t* = 4.73 CI = [0.09–0.22]). In hypothesizing H4, we find that BI and BA showed a positive and significant serial mediation effect on the relationship between perceived quality and QREW usage intention (β = 0.06, *t* = 3.44 CI = [0.03–0.10]). The coefficient

intervals reported for all of the mediation hypotheses do not straddle a zero in between the lower-level and upper-level thresholds, indicating significant mediation effects. In this regard, H2, H3, and H4 are empirically supported. Table 4 displays the results of these direct and indirect effects. Meanwhile, Figure 2 depicts the direct paths among the constructs within the PLS structural model.

**Table 4.** Hypothesis testing.

| | Hypothesized Paths | β | S.E. | *t*-Value | Bias Corrected LLCI | ULCI | Remarks |
|---|---|---|---|---|---|---|---|
| H1 | BA → INT | 0.396 | 0.052 | 7.623 | 0.290 | 0.493 | Supported |
| H2 | PQ → BA → NT | 0.060 | 0.024 | 2.465 | 0.017 | 0.112 | Supported |
| H3 | BI → BA → INT | 0.201 | 0.035 | 5.718 | 0.134 | 0.271 | Supported |
| H4 | PQ → BI → BA → INT | 0.061 | 0.018 | 3.440 | 0.031 | 0.098 | Supported |

Note: PQ = perceived quality; BI = brand image; BA = brand awareness; INT = usage intention; LLCI = lower-level confidence interval; ULCI = upper-level confidence interval.

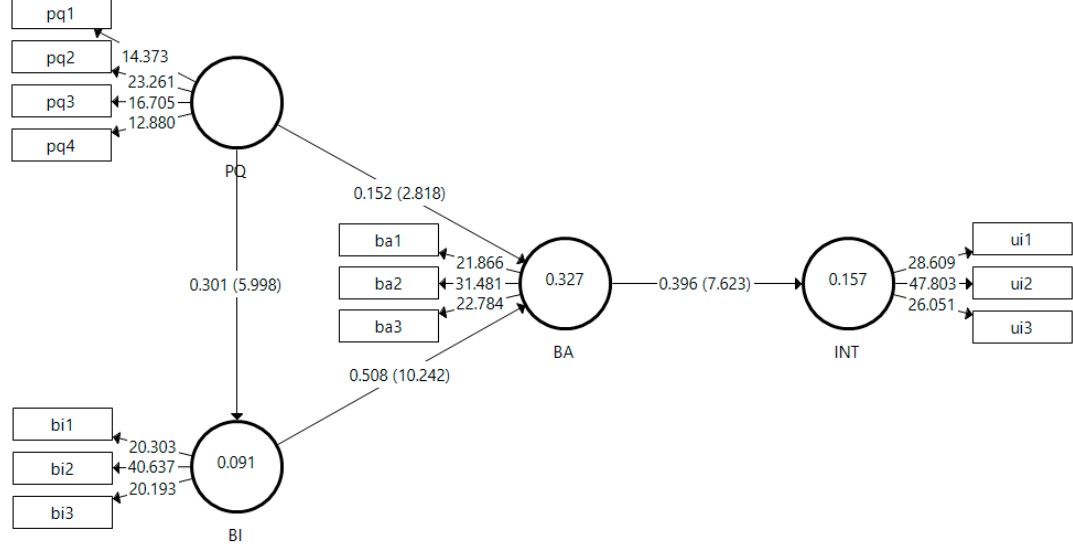

**Figure 2.** PLS structural model.

## 5. Discussion

The interrelationships between brand equity components (perceived value, brand image, and brand awareness) and their impact on consumer intention to use QREWs in the OTC retail environment are examined in this study. Additionally, the investigation into indirect brand equity pathways as a useful tool for modeling users' technology use from the standpoint of the consumer–brand connection is also examined. The current study's PLS path-modeling method demonstrates that brand awareness is the primary factor that significantly affects consumers' intentions to use QREWs in the context of OTC retail transactions (H1). The major findings suggest that E-wallet service providers offer the necessary characteristics and functions demanded by consumers, albeit with different in-app features. Certainly, consumers who find a QREW brand that is more easily recalled compared to other brands (e.g., due to its associated convenience, and practical and emotional values) will most likely consider using the brand when shopping offline [40].

Additionally, consumers' intentions to use QREWs are highly influenced by perceived quality via brand awareness (H2). The result implies that perceived quality can enhance consumers' favorable judgment of QREW brands, which eventually leads to continuous use and brand loyalty. The preceding studies have proven that perceived quality is an important feature that shapes consumers' satisfactory perception of E-wallet services provided that they deliver according to the intended purpose [40]. Similarly, per-

ceived quality and brand awareness are able to develop the brand image, which ultimately yields brand loyalty. The results from the current study are analogous to those reported by [57,58]. The significant findings show that QREW customers are particular about the quality of the QREW services (i.e., dependable and trustworthy) when making a cashless payment decision.

Furthermore, brand image indirectly influences consumers' QREW usage intention via brand awareness (H3). The findings signify that, in addition to its capabilities, which can streamline the payment process toward cashless transactions, consumers' QREW preference is attributable to the effortless recognition of the QREW's brand image. Due to the well-known reputation of the mainstream E-wallet brands, consumers are more familiar with the brands that offer them the most benefits and values, which enables them to make better purchase decisions and, in turn, gain their trust. Similar to previous research, our findings indicate that brand image and brand awareness are the core drivers encouraging consumers' QREW adoption [59,60]. Hence, consumers should not encounter much difficulty in deciding which E-wallet service to choose from based on their visualizations of the tangible cues of the QREW (e.g., logo, interface, theme, etc.) and intangible user experiences.

Perceived quality is the next important factor that significantly and indirectly affects consumers' intentions to use QREWs through brand image and brand awareness (H4). The finding implies that OTC retail consumers are more likely to use QREWs when their impression of the perceived quality of a QREW is high. The good service quality of mobile commerce is often exemplified by reliability, personalized user experience, and trustworthiness [61]. In addition to aesthetic qualities, a positive user experience resulting from these attributes helps consumers generate favorable mental imagery of a brand, which ultimately results in greater brand recognition. Consequently, consumers may evaluate the long-term benefits of the quality services of a QREW after comparing them to other QREW alternatives, thereby boosting their brand loyalty, which is attributable to their familiarity with and appreciation for that particular QREW brand [62].

### 5.1. Theoretical Implications

Our study offers theoretical contributions as follows. First, we contribute to the current literature by developing and testing a model based on the brand equity paradigm to explore the influence of its components on consumers' QREW usage behavior. Most previous research on brand equity and E-wallets has investigated this outcome based on technology acceptance-based frameworks (e.g., TAM and UTAUT). Although this consumer-brand relationship approach has been previously investigated in mobile service adoption [8], this type of modeling is rarely seen in E-wallet adoption except for a few recent studies [7,33].

Notwithstanding, in these studies, brand equity was positioned as a second-order construct [7]. Since incorporating a single composite construct may limit the understanding of the potential meaningful individual effects of the multiple first-order constructs, the disaggregated approach employed by this study shall address this issue [63]. Furthermore, configuring brand equity elements as interconnected antecedents in the model allows researchers to comprehend how these variables are related to each other, which was previously modeled as endogenous variables [33].

Second, by investigating the serial mediating impacts of brand image and brand awareness, this study revealed the underlying mechanism that could explain the effect of perceived quality on consumers' QREW adoption. This research provides a disaggregated solution to the question of how CBBE factors influence consumers' adoption of QREWs. Empirically, the PLS path modeling method reveals that all the studied components had a significant impact on consumers' intentions to use QREWs in the OTC retail environment via both single and dual-mediation pathways. These findings complement the existing literature on mobile payments by focusing exclusively on QREW apps in consumer-facing and proximity-oriented retail settings. In particular, despite the fact that perceived quality is the prime predictor of customers' intentions to use QREWs, brand image, and brand awareness produce positive interventions in spurring this outcome.

Thirdly, the findings of the study contextualize QR-code E-wallets in over-the-counter retail transactions, where technological adoption is in its infancy. Moreover, rather than considering the QREW payment solution as a rival to an NFC-based mobile payment system, it complements the digital payment ecosystem, particularly in markets where hardware- and system-related costs appear to be the primary hurdles to the widespread adoption of E-wallets. Theoretically, this study contributes to a better understanding of consumers' QREW adoption from the perspective of the consumer-brand relationship, which has been the subject of relatively little prior research [7,64], as opposed to the technology acceptance-based frameworks that dominate the mobile payments literature.

*5.2. Practical Implications*

This study has several practical implications that E-wallet service providers can gain from the empirical evidence. Unquestionably, E-wallet service providers should expend more effort toward QREW acceptance by enhancing their apps' features and user interfaces since these physical and design-oriented factors frequently influence the customers' first impressions [65]. Attractive design elements can positively impact the perceived quality of QREW apps, which in turn, can project a positive brand image among users. This can be achieved by incorporating consistency in design, ease of navigation, attention to detail, intuitive layouts, '*mobile-first*' design, interactive elements, and personalization. These elements can help to create a sense of familiarity and continuity, render the app more user-friendly and easier to navigate and create a strong visual identity for the app.

From the perspective of the consumer-brand relationship, a strong brand identity enhances consumers' overall shopping experience. In this context, E-wallet service providers should scrutinize the barriers toward QREW adoption and facilitate its implementation in the entire retail chain network through better information technology integration and effective QREW knowledge transfer [32]. E-wallet service providers should re-evaluate their operational capabilities to enable consumers to rely on QREW apps securely and conveniently for even the most minor transactions. In order to entice them to substitute cash with QREW solutions in the OTC retail environment, the transactions should be highly secure and hassle-free [66].

Similarly, E-wallet service providers should devote effort and time to assessing the long-term branding implications of QREW offerings. QREW service providers must develop a clear branding strategy by consistently delivering high-quality services. This includes factors such as reliability, trustworthiness, and ease of use, which are critical for shaping consumers' perceptions of the service's overall quality. By prioritizing these factors, QREW providers can establish a positive brand image, which can lead to increased brand awareness and adoption of their services. Additionally, in order to increase cashless transaction adoption, policymakers and E-wallet service providers must strive to transform QREW service delivery to be more sustainable and accessible to consumers and retailers [67]. The underprivileged segment of society, such as microretailers, petty traders, and street hawkers, could greatly benefit from this digital transformation. The participation of these unbanked and underserved groups shall address the issue of financial inclusivity among the public [9]. For such efforts to materialize, there needs to be a continuous dialogue among QREW providers, government-related agencies, consumer advocates, and other stakeholders. Among the core agenda items of this dialogue are to improve QR-code payment interoperability among the financial and non-financial institution providers and establish a dedicated "*watchdog*" agency that regulates matters related to the rights and interests of consumers and small traders.

Additionally, in realizing the importance of QREWs to enhance and improve the growth of the retail industry, E-wallet service providers should formulate benchmarking strategies for other similar technology-driven services. This includes benchmarking brand awareness, image, and service quality to identify areas for improvement. Benchmarking can also help providers understand their competitors' strengths and weaknesses, allowing them to differentiate their service offerings and improve the overall value proposition.

Besides, by benchmarking their service quality, E-wallet service providers can identify areas where they need to improve the delivery of their service to meet and exceed the expectations of their consumers. This can result in a more positive brand image, an increase in brand recognition, and a more successful adoption of their QREW services.

*5.3. Limitations and Future Research Directions*

Future research is recommended to adapt the current research framework in a different setting and conduct a comparison encompassing several countries. Moreover, the sample size could be expanded to cover different geographical areas and involve a wider market segmentation. By expanding the sample size that covers different geographical areas, the generalizability of the results could be further improved. Additionally, future research is suggested to examine any additional constructs that fall under the technology-based financial service adoption outside of the well-established technology acceptance domain to strengthen the variance explained in the study. Besides, future research may contemplate a mix of quantitative and qualitative techniques to examine the relationship among the constructs and further explore the strength of the relationships. The valuable insights from this study could be taken forward in the motivation of future research focusing on comparative studies and longitudinal studies of consumer intention with QREW adoption within the OTC retail environment in different countries (i.e., developed and developing). Next, the research was carried out in the greater Klang Valley, Malaysia, where this is the only study to have been carried out based on this research framework. Thus, the findings may not be generalizable to the country's entire population or other countries with different socioeconomic statuses. Future studies may focus on conducting a multicountry comparison of the interrelationships between brand equity components and their influence on consumers' QREW usage intention in the OTC retail context. Moreover, different configurations of indirect pathways involving brand equity elements and integration with other marketing-related theories (e.g., social identity and use gratification) can be considered for modeling users' technology utilization from the perspective of the consumer–brand relationship.

## 6. Conclusions

The study contributes to the current literature on brand equity and E-wallet adoption by exploring the influence of brand equity components on consumers' QREW usage behavior via mediation and serial mediation approaches. The results show that all the studied components have a significant impact on consumer intention to use QREWs in the OTC retail environment. This research provides a disaggregated solution to the question of how CBBE factors influence consumers' adoption of QREWs in over-the-counter retail transactions, where technological adoption is in its infancy. From a practical perspective, E-wallet service providers can gain insights into enhancing their apps' features and user interfaces to positively impact the perceived quality of QREW apps and create a sense of familiarity and continuity, render the app more user-friendly, and create a strong visual identity for the app. E-wallet service providers should scrutinize the image-related barriers toward QREW adoption and facilitate its implementation in the entire retail chain network.

Beyond the OTC retail context, the integration of E-wallet services with other digital services, such as e-commerce and e-government platforms, has made it possible for users to complete transactions effortlessly, resulting in a more streamlined and efficient experience. It is apparent this digital innovation will continue to play a key role in the future of the fintech and the banking industries, given the constant improvement in technology and the rising demand for digital services [68,69]. As a result, banks must invest in the development and integration of digital payment solutions to satisfy the changing demands and expectations of clients. E-wallet services have revolutionized the way customers interact with their finances, and their influence will continue to be felt in the coming years.

**Author Contributions:** Conceptualization, M.I.H. and F.A.A.R.; methodology, M.I.H. and F.A.A.R.; formal analysis, M.I.H.; investigation, F.A.A.R.; data curation, M.I.H. and F.A.A.R.; writing—original draft preparation, F.A.A.R., M.I.H., S.N.W. and R.S.; writing—review and editing, M.I.H.; visualization M.I.H.; supervision, M.I.H.; project administration, M.I.H.; funding acquisition, M.I.H. All authors have read and agreed to the published version of the manuscript.

**Funding:** This research was funded by the Office of the Deputy Vice Chancellor (Research and Innovation), UNIVERSITI TEKNOLOGI MARA, MALAYSIA, via the *Geran Inisiatif Penyeliaan*; grant number 600-RMC/GIP 5/3 (059/2022).

**Institutional Review Board Statement:** Ethical review and approval were not required for this study as it adhered to the institutional requirements in place at the time of the research.

**Informed Consent Statement:** Informed consent was obtained from all subjects involved in the study.

**Data Availability Statement:** The data presented in this study are available on request from the corresponding author. The data are not publicly available due to undisclosed reasons.

**Acknowledgments:** The authors wish to express their gratitude to the Office of the Deputy Vice-Chancellor (Research and Innovation) at Universiti Teknologi MARA Malaysia for their administrative and financial support, which was instrumental in completing this research.

**Conflicts of Interest:** The authors declare no conflict of interest. The funders had no role in the design of the study; in the collection, analyses, or interpretation of data; in the writing of the manuscript; or in the decision to publish the results.

## Appendix A

**Table A1.** Loadings and crossloadings.

|  | PQ | BI | BA | INT |
|---|---|---|---|---|
| PQ1 | *0.730* | 0.273 | 0.248 | 0.217 |
| PQ2 | *0.797* | 0.247 | 0.225 | 0.224 |
| PQ3 | *0.732* | 0.179 | 0.233 | 0.221 |
| PQ4 | *0.696* | 0.170 | 0.185 | 0.234 |
| BI1 | 0.231 | *0.760* | 0.439 | 0.196 |
| BI2 | 0.289 | *0.856* | 0.430 | 0.260 |
| BI3 | 0.200 | *0.780* | 0.459 | 0.239 |
| BA1 | 0.190 | 0.450 | *0.738* | 0.359 |
| BA2 | 0.227 | 0.421 | *0.825* | 0.284 |
| BA3 | 0.305 | 0.438 | *0.808* | 0.291 |
| UI1 | 0.294 | 0.249 | 0.341 | *0.837* |
| UI2 | 0.224 | 0.253 | 0.354 | *0.870* |
| UI3 | 0.242 | 0.229 | 0.301 | *0.815* |

Note: PQ = perceived quality; BI = brand image; BA = brand awareness; INT = usage intention.

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
