# Peer review of "Modeling the Brand Equity and Usage Intention of QR-Code E-Wallets"

_fintech, doi:10.3390/fintech2020013_

Round 1
Reviewer 1 Report
Please see the attached file.

Author Response
Author’s response to Referees’ comments
Reviewer #1 Comments
Thank you for your insightful remarks and comments. Based on these suggestions, we have revised our manuscript, and our responses are shown below, point by point.
- Please delete the context in Line 30-38. It looks like the instruction of paper writing for this journal.
Authors’ response:
Thank you for the observation. We apologize for the error and we have removed the content from the original template.
- Please delete the context in Line 30-38. It looks like the instruction of paper writing for this journal. The authors miss a lot of citations for their context. Please examine the sections of Introduction and Literature review add the sources properly. I mention a few as follow.
Line 39-46; Line 58-64; Line 65-74; Line 75-85; Line 118-128; Line 142-147
Authors’ response:
Thank you for pointing these out. Since the first paragraph is being removed (refer to comment no.1), and some content revisions (in response to other comments), there line indicators have slightly shifted. (a) For Line 38-48, two citations were included; (b) For Line 61-66, we included two citations; (c) For Line 69-79, three citations were added; (d) For Line 78-88, four sources were cited; (e) For Line 110-120; three citations were included (f) For Line 132-138, four sources were cited.
- It looks unusual of the way presenting the hypotheses H2-H4. In general, I would say, for example, perceived quality positively influences brand awareness.
Authors’ response:
Thank you for your comment. Allow us to clarify. The statements in H2 until H4 reflect the format of the mediating hypotheses.
A sample of a paper that utilizes this hypothesis format for direct and serial mediation analyses is exemplified in these papers:
https://link.springer.com/article/10.1007/s11356-022-24899-1
https://www.frontiersin.org/articles/10.3389/fpsyg.2021.743936/full
- Based on aforementioned comment, the authors have to conduct an additional analysis to identify the mediation effects of the variables toward QREW usage intention. You may refer to the following articles:
Baron, R.M.; Kenny, D.A. The moderator–mediator variable distinction in social psychological research: Conceptual, strategic, and statistical considerations. J. Pers. Soc. Psychol. 1986, 51, 1173.
Hair, J.; Black, W.; Babin, B.; Anderson, R. Multivariate Data Analysis, 7th ed.; Pearson Prentice Hall: Upper Saddle River, NJ, USA, 2010.
Authors’ response:
The Baron and Kenny's (1986) mediation method (causal step approach) is no longer considered an appropriate approach for mediation analysis in quantitative marketing research. The more contemporary and sophisticated bootstrapped indirect effect model proposed by Professor Andrew Hayes has replaced the Baron and Kenny's method as the preferred approach for mediation analysis (Hayes, 2009; Preacher & Hayes, 2008). Furthermore, Professor Joseph Hair acknowledged this bootstrapped mediation approach in his PLS modelling book (Hair et al., 2017).
Hayes, A. F. (2009). Beyond Baron and Kenny: Statistical mediation analysis in the new millennium. Communication monographs, 76(4), 408-420.
Preacher, K. J., & Hayes, A. F. (2008). Asymptotic and resampling strategies for assessing and comparing indirect effects in multiple mediator models. Behavior Research Methods, 40(3), 879–891. https://doi.org/10.3758/BRM.40.3.879
Hair, J., Hult, G. T. M., Ringle, C., & Sarstedt, M. (2017). A Primer on Partial Least Squares Structural Equation Modeling (PLS-SEM) (2nd ed.). Sage Publications. https://doi.org/10.1016/j.lrp.2013.01.002
- Referring to Comment 3, Table 4 is the results of testing the mediating effects. I would like to suggest the authors adding the estimation results regarding the new (correct) model.
Authors’ response:
Hayes (2009) dismissed the older method of estimating individual paths in a mediation model (a, b, c’) in order to assess whether a variable M functions as a mediator of the relationship between X and Y. Similarly, Hayes (2009) argued that Baron and Kenny's (1986) mediation method are no longer relevant due to several reasons. Firstly, the method requires the existence of a significant relationship between the independent variable and the dependent variable, which may not be necessary for mediation analysis. Secondly, the method ignores the role of measurement error, which can lead to biased estimates. Thirdly, the method assumes that the mediator is a single construct, which may not be appropriate in complex models with multiple mediators. Fourthly, the method does not account for the indirect effects of multiple mediators, which can result in incomplete or misleading conclusions.
- Please add the information about Line 548-596. Looks like the authors rush to submit this article and miss a lot of context required.
Authors’ response:
Thank you for your meticulous observation. We have included the information on the supplementary materials, author contributions, funding, acknowledgement and other required statements.
Reviewer 2 Report
The authors present the results of a survey in Malaysia. They analyze a special payment method (QR-code E wallets). They do not investigate the advantages of this method compared to other methods. The survey is responded by persons who use the QR-code E-wallets. In this sense the results are biased.
Page 1, lines 30-38: Please skip this paragraph. It is comment of a reviewer.
Page 1, line 44: m-payment and e-money is not the same: If you think please mention the definitions and the quotes.
Page 2 line 63: Please explain TAM and UTAUT. 37-38:
Page 2, line 78: Please present relevant data over more than three years. Absolute and relative values.
Page 3, line 99: The authors present the results of a survey. The survey does not include elements which models a causal relationship in the sense of modern econometrics.
Page 3, lines 118 to 128: This is a literature review without a quote. Please present the relevant literature for your statements.
Page 5, line 207: The hypothesis is complicated. It reflects the intention of a marketing model in specifying a SEM relationship.
Page 6, line 263: It is really complicate to identify such a relationship.
Page 6, lines 277-284: How are the respondents selected? When are the questionnaires distributed? There is a bias in the sample. The authors offer the questionnaire only to people who use the QR-type E-wallet. The results do not represent all visitors of the malls and the results are useless in relation to the whole population of Malaysia. Please offer the questionnaire upon request. Please give a hint that a lot of questions are given in Tables 1 and 2.
Page 7, lines 296-310: Please make clear the response of different question are aggregated to a item, which are use to test the hypotheses.
Page 7, lines 313-323: The results indicate that the survey is not representative for the whole country. It is relevant for a special group who may influence the development of the country.
Page 7, line 332: Please change lesat in least.
Page 8, line 352: Please change R2 into R².
Page 9, Chapter 5: The chapter is really long. The main paper does not include anything about costs. It does not give any numbers. The paper does not describe the technical part of the payment method. Therefore, the chapter should focus on the main results of the paper and the main conclusions.
Page 13; lines 548 to 596: Please give the relevant information and not the general hints of the journal.
Author Response
Author’s response to Reviewers’ comments
Reviewer #2 Comments
- The authors present the results of a survey in Malaysia. They analyze a special payment method (QR-code E wallets). They do not investigate the advantages of this method compared to other methods. The survey is responded by persons who use the QR-code E-wallets. In this sense the results are biased.
Authors’ response:
We appreciate your concerns on this matter.
In consumer marketing research that incorporates brand equity within the model, the consumers must have experience in using the particular brand. Yoo, Donthu and Lee (2000) remarked that “If respondents have known and experienced the products well, they would be able to provide reliable and valid responses to the questionnaire.”. In addition to that, we have clarified that in order to be eligible to participate in the survey, participants had to be at least 18 years old and had used a QR-type E-wallet at least once within the previous three months. This would avoid the issue of heterogeneity (bias) that potential causes error in the model’s estimation.
Yoo, B., Donthu, N., & Lee, S. (2000). An examination of selected marketing mix elements and brand equity. Journal of the Academy of Marketing Science, 28(2), 195–211. https://doi.org/10.1177/0092070300282002
In terms of why QR-code E-wallets are chosen (rather than NFC type), we have justified this context in the article (page 1; 1st paragraph), as follows:
In Malaysia, there are 53 different E-wallet options, making up a significant portion (19%) of the local Fintech industry [2]. A majority of these offerings capitalize on the QR-based E-wallet (QREW) format as opposed to the near-field communications (NFC) interface due to its cheaper implementation for both merchants and customers. Due to the high levels of competition in this industry, there is now an excess of QREW services on the market, all vying for the same share of a niche that is already well fulfilled by credit and debit cards [3].
Besides, QR-code E-wallet serves as the preferred mobile payment method among consumers in many emerging markets, hence making it a promising avenue for future E-wallets adoption research.
- Page 1, lines 30-38: Please skip this paragraph. It is comment of a reviewer.
Authors’ response:
Thank you for the observation. We apologize for the error and we have removed the content from the original template.
- Page 1, line 44: m-payment and e-money is not the same: If you think please mention the definitions and the quotes.
Authors’ response:
Thank you for highlighting this. We have removed the word ‘e-money’ from the text (page 1: 1st paragraph).
- Page 2 line 63: Please explain TAM and UTAUT. 37-38:
Authors’ response:
Thank you for your observation. We have explained the full names of TAM and UTAUT in the text (page 2: 2nd paragraph).
- Page 2, line 78: Please present relevant data over more than three years. Absolute and relative values.
Authors’ response:
Thank you for making this point. We have updated the content with recent and relevant references (page 2, 4th paragraph), as follows:
In these markets, QREWs have gained more popularity than NFC-based E-wallets for economic reasons [9]. The NFC interface has the disadvantages of requiring a membership to bank services and expensive hardware investments (e.g. terminals for merchants, and NFC-compatible smartphones for consumers) [1]. Inadvertently, these elements have made QREW an attractive OTC mobile payment option, given that it just requires a minimal setup consisting of QREW app/service registration, a camera-equipped smartphone, and a QR-code print display [10].
- Page 3, line 99: The authors present the results of a survey. The survey does not include elements which models a causal relationship in the sense of modern econometrics.
Authors’ response:
We appreciate your concern on this matter. Please allow us to offer our views. A psychometric-oriented consumer marketing study and an econometrics research differ in some ways. One key difference between consumer marketing research and econometrics approaches is the type of data used in the model estimation process. Marketing studies, as exemplified by our work, often use primary data collected through surveys that typically involve latent variables, such as psychometric or subjective measurements (Malhotra et al., 2019). These latent variables are not directly observable but are inferred from respondents' responses to survey questions. In contrast, econometrics typically relies on secondary or panel data, which are mostly ‘stand-alone’ and observed variables, such as price, quantity, and demographic information (income, age, gender, etc.) (Wooldridge, 2019). These data are often obtained from secondary sources that collect and publish data on various economic indicators.
Another key difference is the analytical approach used to analyze the data. Marketing studies often employ structural equation modeling (SEM) or other advanced statistical techniques to estimate the causal relationships among latent variables. Econometric analysis, on the other hand, often involves regression analysis or other more traditional statistical methods to estimate the relationship between observed variables.
The SEM model provides an estimate of the strength and direction of the causal relationships between the latent variables and other variables in the model, with primary focus on maximization of the variance explained in the endogenous (outcome) variables (Hair et al., 2017). This is different from the econometric approach, which focus on building formulas and equations to predict the hypothesized causal relationships.
Overall, while both psychometric (consumer marketing surveys) and econometrics approaches aim to understand and explain individuals’ behavior, they differ in their approach to data collection, the types of variables used in analysis, and the analytical methods employed (Goldberger, 1971).
Malhotra, Nunan & Birks (2020). Marketing Research: Applied Insight. Pearson.
Wooldridge, J. M. (2019). Introductory Econometrics: A Modern Approach. Cengage Learning.
Goldberger, A. S. (1971). Econometrics and psychometrics: A survey of communalities. Psychometrika, 36(2), 83-107.
Hair, J., Hult, G. T. M., Ringle, C., & Sarstedt, M. (2017). A Primer on Partial Least Squares Structural Equation Modeling (PLS-SEM) (2nd ed.). Sage Publications. https://doi.org/10.1016/j.lrp.2013.01.002
- Page 3, lines 118 to 128: This is a literature review without a quote. Please present the relevant literature for your statements.
Authors’ response:
Thank you for your observation. We have included three relevant sources (page 3, 2nd paragraph).
- Page 5, line 207: The hypothesis is complicated. It reflects the intention of a marketing model in specifying a SEM relationship.
Authors’ response:
Thank you for your comment. Allow us to clarify. The statements in H2 until H4 reflect the format of the mediating hypotheses.
Below is a sample of papers that utilize this hypothesis format for direct and serial mediation analyses:
https://link.springer.com/article/10.1007/s11356-022-24899-1
https://www.sciencedirect.com/science/article/pii/S2199853122011416
https://www.emerald.com/insight/content/doi/10.1108/IJBM-01-2019-0034/full/html
- Page 6, line 263: It is really complicate to identify such a relationship.
Authors’ response:
Thank you for your comment. Our answer is similar to the above, and the serial mediation relationship is exemplified in the source articles in point 8 above.
- Page 6, lines 277-284: How are the respondents selected? When are the questionnaires distributed? There is a bias in the sample. The authors offer the questionnaire only to people who use the QR-type E-wallet. The results do not represent all visitors of the malls and the results are useless in relation to the whole population of Malaysia. Please offer the questionnaire upon request. Please give a hint that a lot of questions are given in Tables 1 and 2.
Authors’ response:
Thank you for your comments. The mall intercept method was employed. The survey was conducted in June 2021 and took around four months to complete. The following sentences reflect the revised content.
The questionnaire is filled out by the respondents at the exit points of these shopping locations. The mall intercept method provides quick, spontaneous, and honest answers since the respondents' memories about their shopping experience are fresh [42].
Thank you for your concern on the samples’ biasness. However, in measuring brand equity, the consumers must have experience in using the particular brand. Yoo, Donthu and Lee (2000) remarked that “If respondents have known and experienced the products well, they would be able to provide reliable and valid responses to the questionnaire.”. In addition to that, we have clarified that in order to be eligible to participate in the survey, participants had to be at least 18 years old and had used a QR-type E-wallet at least once within the previous three months. This would avoid the issue of heterogeneity that potential causes error in the model’s estimation.
As for providing a hint of where to look for the questions, we have included the following sentence:
“Table 2 summarizes the questionnaire items for the variables studied.”
The revised content are reflected in page 6 (3rd and 4th paragraph).
Yoo, B., Donthu, N., & Lee, S. (2000). An examination of selected marketing mix elements and brand equity. Journal of the Academy of Marketing Science, 28(2), 195–211. https://doi.org/10.1177/0092070300282002
- Page 7, lines 296-310: Please make clear the response of different question are aggregated to a item, which are use to test the hypotheses.
Authors’ response:
Thank you for the comment. We have address this concern by including the following in section 3.3:
“Since the latent variables are not directly observable, the indicator (questionnaire item) scores were aggregated to reduce the measurement error and provide a more accurate estimation of the SEM model parameters [46].”
- Page 7, lines 313-323: The results indicate that the survey is not representative for the whole country. It is relevant for a special group who may influence the development of the country.
Authors’ response:
Thank you for the concern. We acknowledge that our result may not represent the actual outcome should the study is replicated to sample respondents from the whole country. Thus, we have address this issue in the section ‘Limitations and future research directions”.
“Next, the research was carried out in the greater Klang Valley, Malaysia, where this is the only study have been carried out based on this research framework. Thus, the findings may not be generalizable to the country’s entire population or other countries with different socio-economic statuses.”
However, we would like to justify the the relevance of the survey which data collection was limited to the urban areas. This approach allows for the efficient and cost-effective collection of large-scale data from a specific target population (which a sizeable country’s proportion resides in the capital city). By focusing on a specific geographic area, the study can provide insights and recommendations that are tailored to the needs of local businesses and consumers. While the results may not be generalizable to the entire country, they can still provide valuable insights and serve as a basis for further research and analysis.
- Page 7, line 332: Please change lesat in least.
Authors’ response:
Thank you for the meticulous observation and we apologize for the error. We have corrected the typo accordingly (changed to ‘least’).
- Page 8, line 352: Please change R2 into R².
Authors’ response:
Thank you for the neat observation and we apologize for the error. We have corrected by formatting the number ‘2’ as a superscript (changed to R²).
- Page 9, Chapter 5: The chapter is really long. The main paper does not include anything about costs. It does not give any numbers. The paper does not describe the technical part of the payment method. Therefore, the chapter should focus on the main results of the paper and the main conclusions.
Authors’ response:
Thank you for expressing the above concerns. Please allow us to justify. As with most academic research papers, statistics, numbers and figures are only explicated in the findings section, (or chapter 4; results, analysis). Typically, this section may include tables, graphs, and figures to help illustrate the findings.
In contrast, the discussion chapter (5) provides a more interpretive and explanatory analysis of the results. In this chapter, we interpret the results in the context of the existing literature, draw conclusions and implications from the results, and suggest areas for further research. The discussion chapter also address any limitations of the study and provide suggestions for future research. Unlike the results chapter (4), the discussion chapter (5) allows for more subjective analysis and speculation, and it is typically longer and more detailed than the results chapter.
To address the concern on ‘cost’, and the ‘technical part’ of the payment method, our study gathers and analyses psychometric data from respondents using questionnaire survey method rather than secondary, panel or econometric data (e.g. price, spending, usage trends, income, household size). We would like to reiterate our previous answer to point #6, that our research deals with psychometric-based data aggregated into latent variables from a marketing survey procedure, rather than an econometric approach that employs mostly secondary and historical data.
- Page 13; lines 548 to 596: Please give the relevant information and not the general hints of the journal.
Authors’ response:
Thank you for your meticulous observation. We have included the information on the supplementary materials, author contributions, funding, acknowledgement and other required statements.
Reviewer 3 Report
The paper “Modeling brand equity and usage intention of QR-code E-wallets” is a relevant contribution to the literature dedicated to their problematics.
The authors did a good job of examines and modelling brand equity with usage intention of QR-code E-wallets, and they address knowledge void by augmenting brand equity elements to comprehensively analyse consumers' QREW usage intention in the OTC retail environment.
I see the shortcomings of this paper in the introduction, discussion, conclusions and formal.
The introduction should be expanded to include the issue of innovation. Innovations are mentioned in the introduction. Authors are suggested to look into the work of
Loucanova, E., & Olsiakova, M. (2022). Comparison of Innovation in the Electronic Banking Services of the Largest Slovak Banks. In: Marketing and Management of Innovations,13 (4), 1-9 pp.
Parameswar, N., Dhir, S., & Dhir, S. (2017). Banking on innovation, innovation in banking at ICICI bank. Global Business and Organizational Excellence, 36(2), 6-16.
The research analysis is detailed and consistent.
I recommend splitting the discussion and the conclusion. Highlight the practical and academic implication of this study in the conclusion section.
I recommend editing the article on the formal side: supplementary Materials, Author Contributions, Funding, etc. data are not filled in.
Author Response
Author’s response to Reviewers’ comments
Reviewer #3 Comments
- The paper “Modeling brand equity and usage intention of QR-code E-wallets” is a relevant contribution to the literature dedicated to their problematics.
Authors’ response:
Thank you for the positive comment.
- The authors did a good job of examines and modelling brand equity with usage intention of QR-code E-wallets, and they address knowledge void by augmenting brand equity elements to comprehensively analyse consumers' QREW usage intention in the OTC retail environment.
Authors’ response:
Thank you for acknowledging that our research effectively narrows the literature gap.
- I see the shortcomings of this paper in the introduction, discussion, conclusions and formal.
The introduction should be expanded to include the issue of innovation. Innovations are mentioned in the introduction. Authors are suggested to look into the work of
Loucanova, E., & Olsiakova, M. (2022). Comparison of Innovation in the Electronic Banking Services of the Largest Slovak Banks. Marketing and Management of Innovations,13 (4), 1-9 pp.
Parameswar, N., Dhir, S., & Dhir, S. (2017). Banking on innovation, innovation in banking at ICICI bank. Global Business and Organizational Excellence, 36(2), 6-16.
Authors’ response:
Thank you for the above suggestions. We have cited the references above, and the issue of innovation is reiterated in the new Conclusion section (that also addressed your concern in point #6 below).
- The research analysis is detailed and consistent.
Authors’ response:
Thank you for the positive comment.
- I recommend splitting the discussion and the conclusion. Highlight the practical and academic implication of this study in the conclusion section.
Authors’ response:
Thank you for this valuable feedback. We have included a conclusion section (chapter 6) that briefly summarizes the theoretical and practical implications of this study. We have also put forth a statement that e-Wallet services are beneficial in driving productive digital banking innovation to end users (with citations to the recommended sources).
- I recommend editing the article on the formal side: supplementary Materials, Author Contributions, Funding, etc. data are not filled in.
Authors’ response:
Thank you for your meticulous observation. We have included the information on the supplementary materials, author contributions, funding, acknowledgement and other required statements.
Round 2
Reviewer 1 Report
The authors have improved the manuscript.
Please add your responses regarding the mediation effects of the variables toward RQEW usage intention into the context.
Author Response
Author’s response to Reviewers’ comments
Reviewer #1 Comments
Thank you for your insightful remarks and comments. Based on these suggestions, we have revised our manuscript (highlighted in green), and our responses are shown below, point by point.
- Please add your responses regarding the mediation effects of the variables toward QREW usage intention into the context.
Authors’ response:
Thank you for the observation. We apologize for not including the content in the manuscript. We have included our responses (pertaining to the use of Preacher & Hayes’ indirect effect method) in page 8 of the manuscript, as follows:
“To test the mediation hypotheses (H2 to H4), the bootstrapping indirect effect method of Preacher and Hayes [54] was utilized. Preacher and Hayes’s [54] indirect effect approach is currently the default method in testing mediation effects as compared to the older Baron and Kenny's [55] method because the latter is no longer relevant due to its requirement for a significant relationship between the independent and dependent variables, ignorance of measurement error, assumption of a single construct mediator, and failure to account for multiple mediators' indirect effects [56].”
Reviewer 2 Report
The authors react to my comments. Together with the reactions to the comments of the other reviewers the paper is improved. The survey is not representive for the whole population. Nevertheless, it may give some information for the fintech industry.
Author Response
Author’s response to Reviewers’ comments
Reviewer #2 Comments
- The authors react to my comments. Together with the reactions to the comments of the other reviewers the paper is improved. The survey is not representive for the whole population. Nevertheless, it may give some information for the fintech industry.
Authors’ response:
We appreciate your concerns on this matter. We take note on this, and have briefly explain on the generalizability issue in the ‘Limitations and future research directions’ section, in page 11 of the manuscript, as follows:
“Moreover, the sample size could be expanded to cover different geographical areas and involve a wider market segmentation. By expanding the sample size that covers different geographical areas, the generalizability of the results could be further improved.”